# Pick-a-Pic: An Open Dataset of User Preferences for Text-to-Image Generation

**Yuval Kirstain**[τ]    **Adam Polyak**[τ]    **Uriel Singer**

**Shahbuland Matiana**[σ]    **Joe Penna**[σ]    **Omer Levy**[τ]

[τ] Tel Aviv University
[σ] Stability AI
yuval.kirstain@cs.tau.ac.il

## Abstract

The ability to collect a large dataset of human preferences from text-to-image users is usually limited to companies, making such datasets inaccessible to the public. To address this issue, we create a web app that enables text-to-image users to generate images and specify their preferences. Using this web app we build Pick-a-Pic, a large, open dataset of text-to-image prompts and real users' preferences over generated images. We leverage this dataset to train a CLIP-based scoring function, PickScore, which exhibits superhuman performance on the task of predicting human preferences. Then, we test PickScore's ability to perform model evaluation and observe that it correlates better with human rankings than other automatic evaluation metrics. Therefore, we recommend using PickScore for evaluating future text-to-image generation models, and using Pick-a-Pic prompts as a more relevant dataset than MS-COCO. Finally, we demonstrate how PickScore can enhance existing text-to-image models via ranking.[1]

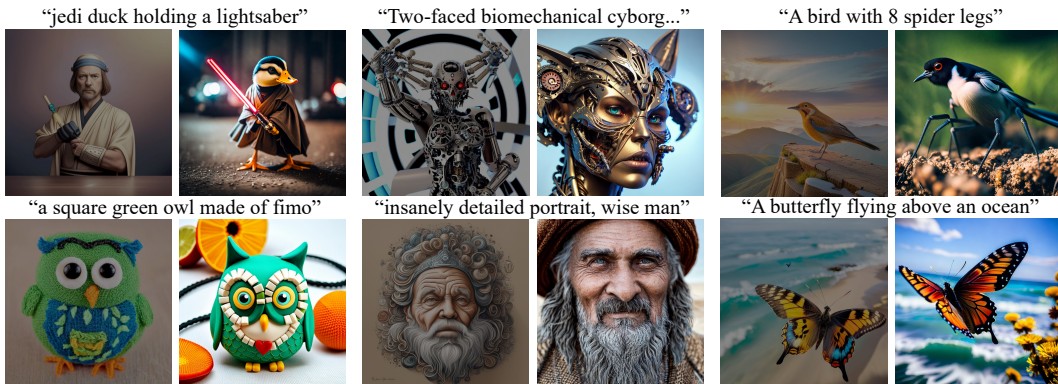

Figure 1: Images generated via our web application, showing darkened non-preferred images (left) and preferred images (right).

---

[1] https://github.com/yuvalkirstain/PickScore

37th Conference on Neural Information Processing Systems (NeurIPS 2023).

# 1 Introduction

Recent advances in aligning language models with user behaviors and expectations have placed a significant emphasis on the ability to model user preferences [10, 1, 3]. However, little attention has been paid to this ability in the realm of text-to-image generation. This lack of attention can largely be attributed to the absence of a *large* and *open* dataset of human preferences over state-of-the-art image generation models.

To fill this void, we create a web application that enables users to generate images using state-of-the-art text-to-image models while specifying their preferences. With explicit consent from the users, we collect their prompts and preferences to create Pick-a-Pic, a publicly available dataset comprising over half-a-million examples of human preferences over model-generated images.[2] Each example in our dataset includes a prompt, two generated images, and a label indicating the preferred image, or a tie when no image is significantly preferred over the other. Notably, Pick-a-Pic was created by *real users* with a genuine interest in generating images. This interest differs from that of crowd workers who lack the intrinsic motivation to produce creative prompts or the original intent of the prompt's author to judge which image better aligns with their needs.

Tapping into authentic user preferences allows us to train a scoring function that estimates the user's satisfaction from a particular generated image given a prompt. To train such a scoring function we finetune CLIP-H [12, 7] using human preference data and an analogous objective to that of InstructGPT's reward model [10]. This objective aims to maximize the probability of a preferred image being picked over an unpreferred one, or even the probability in cases of a tie. We find that the resulting scoring function, PickScore[3], achieves superhuman performance in the task of predicting user preferences (a 70.5% accuracy rate, compared to humans' 68.0%), while zero-shot CLIP-H (60.8%) and the popular aesthetics predictor [14] (56.8%) perform closer to chance (56.8%).

Equipped with a dataset for human preferences and a state-of-the-art scoring function, we propose updating the standard protocol for evaluating text-to-image generation models. First, we suggest that researchers evaluate their text-to-image models using prompts from Pick-a-Pic, which better represent what humans want to generate than mundane captions, such as those found in MS-COCO [2, 9]. Second, to compare PickScore with FID, we conduct a human evaluation study and find that even when evaluated against MS-COCO captions, PickScore exhibits a strong correlation with human preferences (0.917), while ranking with FID yields a negative correlation (-0.900). Importantly, we also compare PickScore with other evaluation metrics using model rankings inferred from real user preferences. We observe that PickScore is more strongly correlated with ground truth rankings, as determined by real users, than other evaluation metrics. Thus, we recommend using PickScore as a more reliable evaluation metric than existing ones.

Finally, we explore how PickScore can improve the quality of vanilla text-to-image models via ranking. To accomplish this, we generate images with different initial random noises as well as different templates (e.g. "breathtaking [prompt]. award-winning, professional, highly detailed") to slightly alter the user prompt. We then test the impact of selecting the top image according to different scoring functions. Our findings indicate that human raters prefer images selected by PickScore more than those selected by CLIP-H [7] (win rate of 71.3%), an aesthetics predictor [14] (win rate of 85.1%), and the vanilla text-to-image model (win rate of 71.4%).

In summary, the presented work addresses a gap in the field of text-to-image generation by creating a large, open, high-quality dataset of human preferences over user-prompted model-generated images. We demonstrate the potential of this dataset by training a scoring function, PickScore, which exhibits a performance superior to any other publicly-available automatic scoring function, in predicting human preferences, evaluating text-to-image models, and improving them via ranking. We encourage the research community to adopt Pick-a-Pic and PickScore as a basis for further advances in text-to-image modeling and incorporating human preferences into the learning process.

---

[2]We recently open-sourced a new version of the Pick-a-Pic dataset that has more than one million examples.
[3]The model is available at `https://huggingface.co/yuvalkirstain/PickScore_v1`

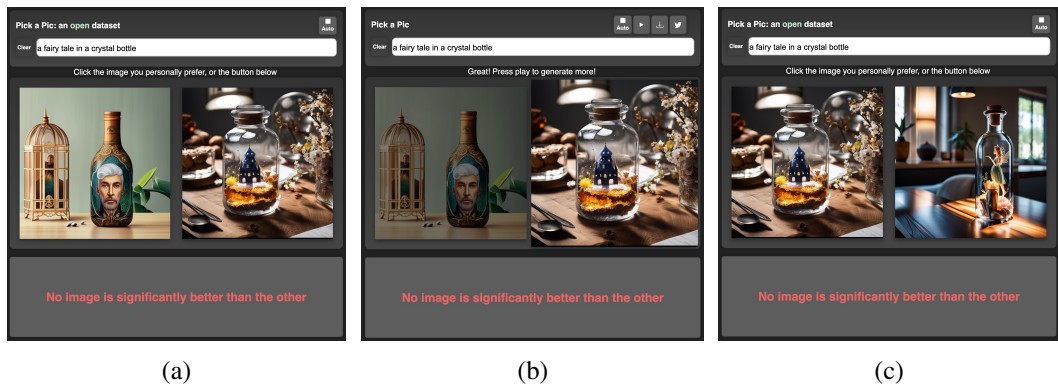

| (a) | (b) | (c) |

Figure 2: How Pick-a-Pic data is collected through the app: (a) the user first writes a caption, and receives two images; (b) the user makes a preference judgment; (c) a new image is presented instead of the rejected image. This flow repeats itself until the user changes the prompt.

## 2 Pick-a-Pic Dataset

The Pick-a-Pic dataset[4] was created by logging user interactions with the Pick-a-Pic web application for text-to-image generation. Overall, the Pick-a-Pic dataset contains over 500,000 examples and 35,000 distinct prompts. Each example contains a prompt, two generated images, and a label for which image is preferred, or if there is a tie when no image is significantly preferred over the other. The images in the dataset were generated by employing multiple backbone models, namely, Stable Diffusion 2.1, Dreamlike Photoreal 2.0[5], and Stable Diffusion XL variants [13] while sampling different classifier-free guidance scale values [6]. As we continue with our efforts to collect more user interactions through the Pick-a-Pic web app and decrease the number of NSFW examples included in the dataset, we will periodically upload new revisions of the dataset.

**The Pick-a-Pic Web App**   To ensure maximum accessibility for a wide range of users, the user interface was designed with simplicity in mind. The application allows users to write creative prompts and generate images. At each turn, the user is presented with two generated images (conditioned on their prompt), and asked to select their preferred option or indicate a tie if they have no strong preference. Upon selection, the rejected (non-preferred) image is replaced with a newly generated image, and the process repeats. The user can also clear or edit the prompt at any time, and the app will generate new images appropriately. Figure 2 illustrates the usage flow.

**Real Data from Real Users**   A key advantage of Pick-a-Pic is that our data is collected from real, intrinsically-motivated users, rather than paid crowd workers. We achieve this by approaching a wide audience through various social media channels such as Twitter, Facebook, Discord, and Reddit. At the same time, we mitigate the risk of collecting low-quality data resulting from potential misuse of the application by implementing several quality control measures. First, users are required to authenticate their identity using either a Gmail or a Discord account.[6] Second, we closely monitor user activity logs and take action to ban users who generate NSFW content, use multiple instances of the web app simultaneously, or make judgments at an unreasonably fast pace. Third, we use a list of NSFW phrases to prevent users from generating harmful content. Last, we limit users to 1000 interactions and periodically increase the limit. These measures work in tandem to ensure the integrity and reliability of Pick-a-Pic's data.

**Annotation Methodology**   While piloting the web app, we experimented with different annotation strategies to optimize for data quality, efficiency of collection, and user experience. Specifically, we tested the following annotation options: (1) 4 images, no ties; (2) 2 images, no ties; (3) 2 images,

---

[4]The dataset is available at `https://huggingface.co/datasets/yuvalkirstain/pickapic_v1`, and its updated version is available at `https://huggingface.co/datasets/yuvalkirstain/pickapic_v2`.

[5]Dreamlike Photoreal 2.0 is a finetuned version of Stable Diffusion 1.5

[6]All users provide explicit consent to share their interactions with Pick-a-Pic's application as part of a public dataset. To ensure privacy, we anonymize user IDs.

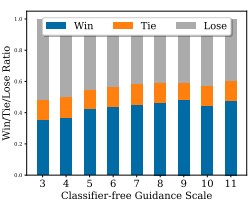
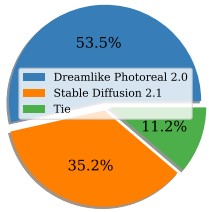

(a) Win rate versus classifier-free guidance scale for Stable Diffusion XL (Alpha).

(b) Preference distribution when comparing Stable Diffusion 2.1 with Dreamlike Photoreal 2.0.

Figure 3: The Pick-a-Pic dataset enables us to perform model selection (a), and model evaluation (b).

with ties. We found that the latter option (2 images, with ties) exceeds the other two in terms of user engagement and inter-rater agreement.

**Preprocessing**    When processing the collected interactions, we filter prompts with NSFW phrases and banned users. We acknowledge that there are still NSFW images and prompts, and will periodically attempt to update the dataset and reduce such occurrences. To divide the dataset into training, validation, and testing subsets, we first sample one thousand prompts, ensuring that each prompt was created by a unique user. Next, we randomly divide those prompts into two sets of equal size to create the validation and test sets. We then sample exactly one example for each prompt to include in these sets. For the training set, we include all examples that do not share a prompt with the validation and test sets. This approach ensures that no split shares prompts with another split, and the validation and test sets do not suffer from being non-proportionally fitted to a specific prompt or user.

**Statistics**    Since the creation of the Pick-a-Pic web app we have gathered 968,965 rankings which originated from 66,798 prompts and 6,394 users. However, as the Pick-a-Pic dataset is constantly updating, the reported experiments in this paper involve an NSFW filtered and not fully updated version of Pick-a-Pic, that contains 583,747 training examples, and 500 validation and test examples. The training set of this dataset contains 37,523 prompts from 4,375 distinct users.

**Model Selection and Evaluation**    The Pick-a-Pic dataset offers a unique opportunity for a model selection and evaluation methodology, leveraging users' preferences for unbiased analysis. To illustrate this opportunity, we use the collected data and analyze the impact of changing the classifier-free guidance scale of Stable Diffusion XL (Alpha variant) on its performance. Specifically, we compare human preferences made when both images were generated by Stable Diffusion XL (Alpha variant) but using different classifier-free guidance scales[7]. For each scale, we compute the win ratio, representing the percentage of judgments where its use led to a preferred image. We also calculate the corresponding tie and lose ratios for each scale, enabling a detailed analysis of which classifier-free guidance scales are more effective. Our results are depicted in Figure 3 (a), and verify for example, that a guidance scale of 9 usually yields preferred images when compared to a guidance scale of 3.

Furthermore, by examining user preferences between images generated by different backbone models, we can determine which model is preferred more by users. For instance, considering judgments in which one image was generated by Dreamlike Photoreal 2.0 and the other by Stable Diffusion 2.1, we can evaluate which model is more performant. As shown in fig. 3 (b), users usually prefer Dreamlike Photoreal 2.0 over Stable Diffusion 2.1. We encourage researchers to contact us and include their text-to-image models in the Pick-a-Pic web app for the purpose of model selection and evaluation.

## 3    PickScore

One valuable outcome from collecting a large, natural dataset of user preferences is that we can use it to train a function that scores the quality of a generated image given a prompt. We train the PickScore scoring function over Pick-a-Pic by combining a CLIP-style model with a variant of

---

[7]The frequency of different guidance scales is similar.

InstructGPT's reward model objective [10]. PickScore is able to predict user preferences in held-out Pick-a-Pic prompts better than any other publicly-available scoring function, surpassing even expert human annotators (Section 4). Such a scoring function can be of value for various scenarios, such as performing model evaluation (Section 5), increasing the quality of generated images via ranking (Section 6), building better large-scale datasets to improve text-to-image models [14], and improving text-to-image models through weak supervision (e.g. RLHF).

**Model**   PickScore follows the architecture of CLIP [12]; given a prompt $x$ and an image $y$, our scoring function $s$ computes a real number by representing $x$ using a transformer text encoder and $y$ using a transformer image encoder as $d$-dimensional vectors, and returning their inner product:

$$s(x, y) = E_{\text{txt}}(x) \cdot E_{\text{img}}(y) \cdot T \tag{1}$$

Where $T$ is the learned scalar temperature parameter of CLIP.

**Objective**   The input for our objective includes a scoring function $s$, a prompt $x$, two images $y_1, y_2$, and a preference distribution vector $p$, which captures the user's preference over the two images. Specifically, $p$ takes a value of $[1, 0]$ if $y_1$ is preferred, $[0, 1]$ if $y_2$ is preferred, or $[0.5, 0.5]$ for ties. Given this input, the objective optimizes the scoring function's parameters by minimizing the KL-divergence between the preference $p$ and the softmax-normalized scores of $y_1$ and $y_2$:

$$\hat{p}_i = \frac{\exp s(x, y_i)}{\sum_{j=1}^2 \exp s(x, y_j)} \tag{2}$$

$$L_{\text{pref}} = \sum_{i=1}^2 p_i \left(\log p_i - \log \hat{p}_i\right) \tag{3}$$

Since many examples can originate from the same prompt, we mitigate the risk of overfitting to a small set of prompts by applying a weighted average when reducing the loss across examples in the batch. Specifically, we weigh each example in the batch, with an inverse proportion to its prompt frequency in the dataset. This objective is analogous to InstructGPT's reward model objective [10]. We also experiment with incorporating in-batch negatives into the objective, but find that this yields a less accurate scoring function (see Appendix).

**Training**   We finetune CLIP-H [7] using our framework[8] on the Pick-a-Pic training set. We train the model for 4,000 steps, with a learning rate of 3e-6, a total batch size of 128, and a warmup period of 500 steps, which follows a linearly decaying learning rate; the experiment is completed in less than an hour with 8 A100 GPUs. We did not perform hyperparameter search, which might further improve results. For model selection, we evaluate the model's accuracy on the validation set (without the option for a tie) in intervals of 100 steps, and keep the best-performing checkpoint.

## 4   Preference Prediction

We first evaluate PickScore on the task it was trained to do: predict human preferences. We find that PickScore outperforms all other baselines, including expert human annotators.

**Metric**   To evaluate the ability of models to predict human preferences, we use an adapted accuracy metric that accounts for the possibility of a tie. Our metric assigns one point to the model for predicting the same label as the user, half a point if either label or prediction is a tie (but not both), and zero otherwise.

**Tie Threshold Selection**   Utilizing the notation specified in Section 3, each model requires a tie threshold probability $t$ to predict a tied outcome when $|\hat{p}_1 - \hat{p}_2| < t$. To achieve this, we evaluate each model on the validation set using various tie threshold probabilities and subsequently determine the most optimal tie threshold for each model. For human experts, we do not perform tie selection and explicitly allow them to select a tie.

---

[8]`https://github.com/yuvalkirstain/PickScore`

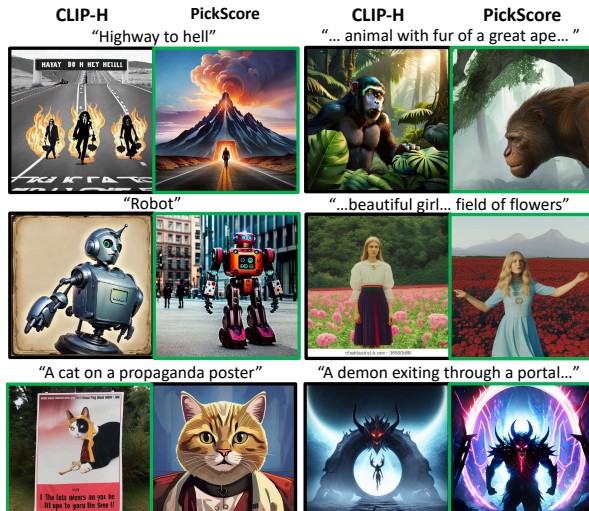

| CLIP-H | PickScore | CLIP-H | PickScore |
|---|---|---|---|
| "Highway to hell" | | "... animal with fur of a great ape... " | |

"Robot"

"...beautiful girl... field of flowers"

"A cat on a propaganda poster"

"A demon exiting through a portal..."

Figure 4: Disagreement between CLIP-H (left) and PickScore (right) on the Pick-a-Pic validation set. We add green borders around images that humans preferred.

Table 1: Quantitative results on Pick-a-Pic.

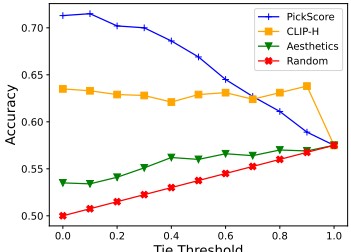

(a) Accuracy across different tie thresholds on the Pick-a-Pic validation set

| Model | Accuracy |
|---|---|
| Random | 56.8 |
| Human Expert | 68.0 |
| Aesthetics [14] | 56.8 |
| CLIP-H [7] | 60.8 |
| ImageReward [18] | 61.1 |
| HPS [17] | 66.7 |
| PickScore (Ours) | **70.5** |

(b) Performance on the Pick-a-Pic test set.

**Baselines**  We compare our model with CLIP-H [7], an aesthetics predictor [14] built on top of CLIP-L [12], a random classifier, and human experts.[9] For completeness, we also compare our results with models from concurrent work, namely, HPS [17] and ImageReward [18].

**Results**  First, we compare the models' performance on the validation set across different tie thresholds. Figure 1a shows that PickScore outperforms the baselines across almost all thresholds, and achieves the highest global score by a wide margin. After selecting the best-performing tie threshold for each model, we use the threshold to evaluate the different models on the test set.

Table 1b shows that the aesthetics score (56.8) and CLIP-H (60.8) perform closer to a random chance baseline (56.8), while PickScore ($70.5 \pm 0.142$)[10] achieves superhuman performance, as it even outperforms human experts (68.0). It is important to emphasize a core difference between *real users* that produce the ground truth labels and *annotators* used to evaluate human performance. The users that produce the ground truth are actual text-to-image users, which have an idea (which may be incomplete) for an image, and invent a prompt (which may lack details) with hope that the resulting image will match their preferences. In contrast, annotators that are used to measure human performance are oblivious to the original user's context, idea, and motivation. Therefore, superhuman performance on this task means that the model is able to outperform a human annotator that is oblivious to the original user's context, idea, and motivation. The superhuman performance of PickScore showcases the importance of using real users as ground truth rather than expert annotators when collecting human preferences. Moreover, the relatively modest human performance (68.0) on the task, when compared to a random baseline (56.8), shows that predicting human preferences in text-to-image generation is a difficult task for human annotators.

For completeness, we also include the results of concurrent work from HPS [17], which scores 66.7, and ImageReward [18], which scores 61.1; PickScore outperforms both. To further illustrate the differences between CLIP-H and PickScore, we showcase examples of disagreement from the Pick-a-Pic validation set in Figure 4. We notice that PickScore often chooses more aesthetically pleasing images than CLIP-H; at times, at the cost of faithfulness to the prompt.

---

[9]Throughout this paper, we use friends and colleagues of the authors for expert human annotation.

[10]We train PickScore using three different random seeds and report the mean and variance.

**Stable Diffusion 2.1**  **Dreamlike Photoreal 2.0**

CFG 3        CFG 9       CFG 3        CFG 9

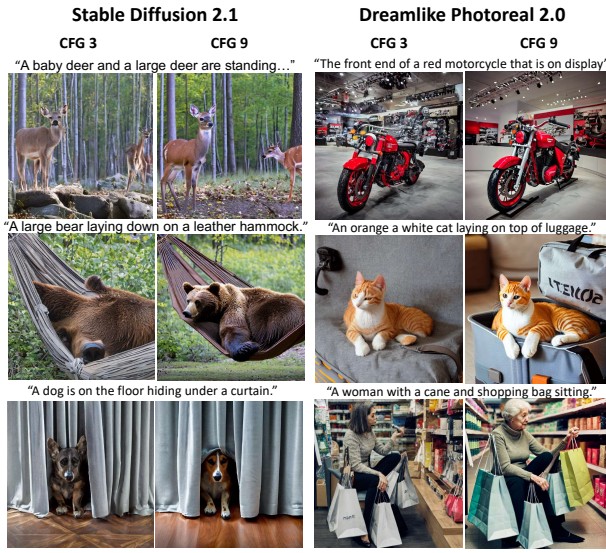

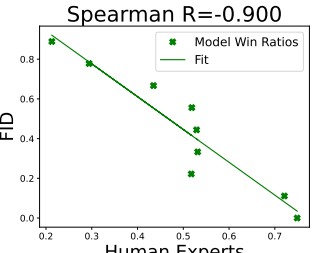

Spearman R=-0.900

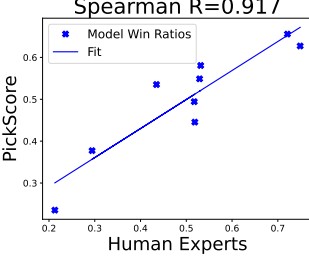

Spearman R=0.917

Figure 5: Images generated using the same seed and model, but using different classifier-free guidance (CFG) scales. Even though high guidance scales lead to worse FID, humans usually find them more pleasing than low guidance scales.

Figure 6: Correlation between the win ratio of different models according to FID and PickScore to human experts on the MS-COCO validation set.

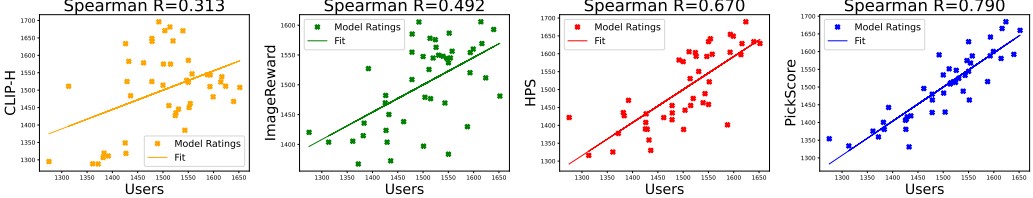

Figure 7: Correlation between Elo ratings of real users and Elo ratings by CLIP-H, ImageReward [18], HPS [17], and PickScore.

## 5 Model Evaluation

Despite significant progress in the generative capabilities of text-to-image models, the standard and most popular prompt dataset has remained the Microsoft Common Objects in Context dataset (MS-COCO) [9]. Similarly, the Fréchet Inception Distance (FID) [5] is still the main metric used for model evaluation. In this section, we explain why we recommend researchers to evaluate their models using prompts from Pick-a-Pic rather than (or at least alongside) MS-COCO, and show that when evaluating state-of-the-art text-to-image models, PickScore is more aligned with human judgments than other automatic evaluation metrics.

**Model Evaluation Prompts**   The prompts contained in the MS-COCO dataset are captions of photographs taken by amateur photographers, depicting objects and humans in everyday settings. While certain captions within this dataset may pose a challenge to text-to-image models, it is evident that the scope of interest for text-to-image users extends beyond commonplace objects and humans. Moreover, the main use-case of image generation is arguably to generate *fiction*, which *cannot* be captured by camera. By construction, Pick-a-Pic's prompts are sampled from real users, and thus better represent the natural distribution of text-to-image intents. We therefore strongly advocate that the research community employ prompts from Pick-a-Pic when assessing the performance of text-to-image models.

**FID**   The FID metric [5] gauges the degree of resemblance between a set of generated images and a set of authentic images, at the set level. To do so, it first embeds the real and generated images into

the feature space of an Inception net [15], and then estimates the mean and covariance of both sets of images and calculates their similarity. FID is thus geared towards measuring the realism of a set of images, but is oblivious to the prompts. In contrast, PickScore provides a per-instance score, and is directly conditioned on the prompt. We thus hypothesize that PickScore will correlate better with human judgements of generation quality.

To empirically test our hypothesis with the most "convenient" settings for the FID metric, we select 100 random captions from MS-COCO validation split. For each caption, we generate images from 9 different models based on the same set of prompts, and ask human experts to rank the 9 generated images (with ties), inducing pairwise preferences. Specifically, we use Stable Diffusion 1.5, Stable Diffusion 2.1, and Dreamlike Photoreal 2.0 combined with three different classifier-free guidance scales (3, 6, and 9). We then repeat the labeling process (over the same images) using FID, and PickScore instead of humans to determine preferences. Since FID does not operate on a per-example base, when "labeling" with FID, we simply choose the model that has a lower (better) FID score on MS-COCO.

Figure 6 shows the correlation between model win rates induced by human rankings (horizontal) and model win rates induced by each automatic scoring function. PickScore exhibits a stronger correlation (0.917) with human raters on MS-COCO captions than FID (-0.900), which surprisingly, exhibits a strong *negative* correlation. As FID is oblivious to the prompt, one would expect zero correlation, and not a strong negative correlation. We hypothesize that this is related to the classifier-free guidance scale hyperparameter – larger scales tend to produce more vivid images (which humans typically prefer), but differ from the distribution of ground truth images in MS-COCO, yielding worse (higher) FID scores. Figure 5 visualizes these differences by presenting pairs of images generated with the same random seed but with different classifier-free guidance (CFG) scales.

**Other Evaluation Metrics**   When comparing with evaluation metrics that do not assume a set of ground truth images, we are able to use a more reliable evaluation procedure. In this evaluation procedure, we consider real user preferences rather than human annotators, as well as more models (i.e. more data points). Specifically, we take all the 14,000 collected preferences that correspond to prompts from the Pick-a-Pic test set. This set of examples contains images generated by 45 different models – four different backbone models, each with different guidance scales. We then use these real user preferences to calculate Elo ratings [4] for the different models. Afterward, we repeat the process while replacing the real user preferences with CLIP-H [7], and PickScore predictions. Similarly to Section 4, we also compare against the concurrent work from ImageReward[18] and HPS [17]. Since the Elo rating system is iterative by nature, we repeat the process 50 times. Each time we randomly shuffle the order of examples, and calculate the correlation with human ratings. Finally, for each metric, we output the mean and standard deviation of its 50 corresponding correlations. Figure 7 displays the correlation between real users' Elo ratings with the different metrics' rating, showing that PickScore exhibits a stronger correlation ($0.790 \pm 0.054$) with real users than all other automatic metrics, namely CLIP-H ($0.313 \pm 0.075$), ImageReward ($0.492 \pm 0.086$), and HPS ($0.670 \pm 0.071$).

## 6   Text-to-Image Ranking

Another possible application for scoring functions is improving the performance of generations made by text-to-image models through ranking: generate a sample of images, and select the one with the highest score. To test this approach, we generate one hundred images for each prompt of the one hundred prompts we take from the Pick-a-Pic test set. We generate the images with Dreamlike Photoreal 2.0, using a classifier-free guidance scale of 7.5, and to increase image diversity, we use 5 initial random noises and 20 different prompt templates. These templates include the null template "`[prompt]`" and other templates like "breathtaking `[prompt]`. award-winning, professional, highly detailed". From each set of 100 generated images, we select the best one according to PickScore, CLIP-H, the aesthetics score, or randomly; in addition, we randomly select one image from the null template for control. We then ask expert human annotators to compare PickScore's chosen image to each of the other functions' choices, and decide which one they prefer.

Table 2 shows that PickScore consistently selects more preferable images than the baselines. By manually analyzing some examples, we find that PickScore typically selects images that are both more aesthetic and better aligned with the prompt. We also measure this explicitly by using the aesthetic scoring function instead of a human rater when comparing PickScore's choice to CLIP-H's,

Table 2: Percentage of instances where humans prefer PickScore's choice over another scoring function's choice when selecting one image out of 100.

| Comparison | Win Rate |
|---|---|
| PickScore vs Random Seed | |
|   + Null Template | 71.4 |
|   + Random Template | 82.0 |
| PickScore vs Aesthetics [14] | 85.1 |
| PickScore vs CLIP-H [7] | 71.3 |

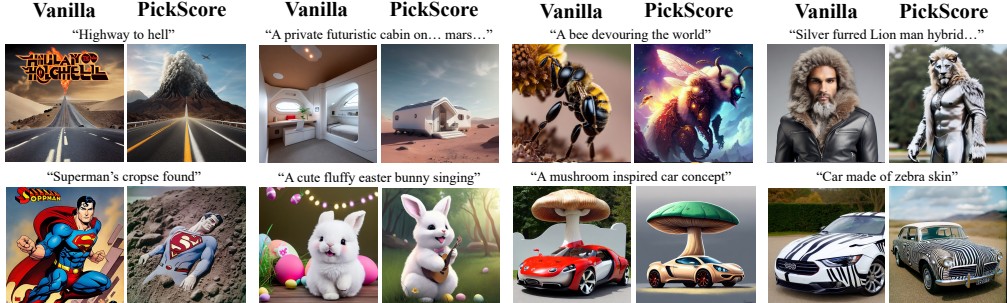

Figure 8: Comparing the image from the vanilla text-to-image model (left) with the image selected by PickScore from a set of 100 generations (right).

and find that for 68.5% of the prompts, PickScore selects an image with a higher aesthetic score. Likewise, when comparing PickScore to the aesthetic scorer, 90.5% of PickScore's choices have a higher CLIP-H text-alignment score than the images chosen by the aesthetic scorer. Figure 8 visualizes the benefits of selecting images with PickScore.

# 7 Related Work

Collecting and learning from human preferences is an active area of research in natural language processing (NLP) [3, 1, 10]. However, in the domain of text-to-image generation, related research questions have received little attention. One notable work that focuses on collecting human judgments in text-to-image generation is the Simulacra Aesthetic Captions (SAC) dataset [11]. This dataset contains almost 200,000 human ratings of generated images. However, unlike Pick-a-Pic which focuses on *general* user preferences, and allows users to compare *between* generated images, the human raters of SAC provide an *absolute* score for the *aesthetic* quality of the generated images.

There has been some concurrent work that involves collection and learning from human preferences that we describe below. Lee et al. [8] consider three simple categories of challenges (count, color, and background), enumerating through templates (e.g. "[number] dogs") to synthetically create prompts. Then, they instruct crowd workers to choose if a generated image is good or bad and use the collected 30,000 examples to train a scoring function. In contrast, Pick-a-Pick allows real users to write any prompts they choose, and provide pairwise comparisons between images that yield more than 500,000 examples.

ImageReward [18] selects prompts and images from the DiffusionDB dataset [16], and collects for them image preference ratings via crowd workers. Since they employ crowd workers that lack the intrinsic motivation to select images that they prefer, the authors define criteria to assess the quality of generated images and instruct crowd workers to follow these criteria when ranking images. They use this strategy to collect 136,892 examples which originate from 8,878 prompts. Importantly, they have not publicly released this dataset. For completeness, we tested ImageReward on the Pick-a-Pic dataset and confirmed that PickScore outperforms it.

Another concurrent work from Wu et al. [17] collects a dataset of human judgments by scraping about 25,000 human ratings (which include about 100,000 images) from the Discord channel of StabilityAI. Similarly to us, they use an objective analogous to that of InstructGPT to train a scoring function that they name Human Preference Score (HPS). As with ImageReward, we evaluated HPS on Pick-a-Pic and saw that PickScore achieves better performance.

The observed superior performance of PickScore over the concurrent work from both HPS and ImageReward on the Pick-a-Pic dataset could be attributed to several factors. For example, differences in implementation (e.g., model size, backbone, hyperparameters), differences in the scales of data, or variations in the distribution of data. Notably, Pick-a-Pic is more than five times larger than the data used to train HPS and ImageReward. Furthermore, ImageReward collects judgments from crowd workers, which may lead to significant differences in data distribution. In contrast, HPS scrapes ratings from the StabilityAI discord channel for real text-to-image users but may be more aligned with this more specific distribution of text-to-image users. We leave Isolating and identifying the specific effects of these factors for future work.

## 8    Limitations and Broader Impact

It is important to acknowledge that despite our efforts to ensure data quality (see section 2), some images and prompts may contain NSFW content that could potentially bias the data, and some users may have made judgments without due care. Moreover, the preferences of users may include biases that may be reflected in the collected data. These limitations may affect the overall quality and reliability of the data collected and should be taken into consideration when considering the broader impact of the dataset. Nonetheless, we believe that the advantages of collecting data from intrinsically-motivated users and publicly releasing it will enable the text-to-image community to better align text-to-image models with human preferences.

## 9    Conclusions

We build a web application that serves text-to-image users and (willingly) collects their preferences. We use the collected data and present the Pick-a-Pic dataset: an open dataset of over half-a-million examples of text-to-image prompts, generated images, and user-labeled preferences. The quantity and quality of the data enables us to train PickScore, a state-of-the-art text-image scoring function, which achieves superhuman performance when predicting user preferences. PickScore aligns better with human judgements than any other publicly-available automatic metric, and together with Pick-a-Pic's natural distribution prompts, enables much more relevant text-to-image model evaluation than existing evaluation standards, such as FID over MS-COCO. Finally, we demonstrate the effectiveness of using our scoring function for selecting images in improving the quality of text-to-image models. There are still many opportunities for building upon Pick-a-Pic and PickScore, such as RLHF and other alignment approaches, and we are excited to see how the research community will utilize this work in the near future.

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

## Appendix

**Comparing Pick-a-Pic Prompts with MS-COCO Captions**

To illustrate the difference between MS-COCO captions and Pick-a-Pic prompts we show prompts from each dataset.

Pick-a-Pic – "forest with ruins, photo", "A panda bear as a mad scientist", "product photo of a sneakers", "photo of a bicycle, detailed, 8k uhd, dslr, high quality, film grain, Fujifilm XT3", "female portrait photo", "Alexander the great, cover art, colorful", "A galactic eldritch squid towering over the planet Earth, stars, galaxies and nebulas in the background...", "Portrait of a giant, fluffy, ninja teddy bear", "insanely detailed portrait, darth vader, shiny, extremely intricate, high res, 8k, award winning", "Giant ice cream cone melting and creating a river through a city".

MS-COCO – "A man with a red helmet on a small moped on a dirt road.", "A woman wearing a net on her head cutting a cake.", "there is a woman that is cutting a white cake", "a little boy wearing headphones and looking at a computer monitor", "A young girl is preparing to blow out her candle.", "A commercial stainless kitchen with a pot of food cooking.", "Two men that are standing in a kitchen.", "A man riding a bike past a train traveling along tracks.", "The pantry door of the small kitchen is closed.", "A man is doing a trick on a skateboard".

**Training a Scoring Function**

We further explored an alternative loss function that is closer to CLIP's original objective. In this loss function, we also incorporate the remaining examples in the batch as in-batch negatives. To elaborate, considering the notation established in section 3 and $y_1^k, y_2^k$ denoting the images corresponding to the $k$-th example in the batch, we formulate $\hat{p}^k$ as follows:

$$\hat{p}_i^k = \frac{\exp s(x, y_i)}{\sum_k \sum_{j=1}^2 \exp s(x, y_j^k)} \tag{4}$$

We anticipated that this objective function would maintain the general capabilities of CLIP with minimal loss in performance. However, our findings demonstrated that PickScore significantly outperforms this objective function, as the latter only produced a scoring function that achieves an accuracy of 65.2 on the Pick-a-Pic test set.

