# OpenReview forum: "Pick-a-Pic: An Open Dataset of User Preferences for Text-to-Image Generation"
_NeurIPS.cc/2023/Conference — NeurIPS 2023 poster_

### Official Review · Reviewer_hbT1 · 2023-06-29

**Soundness:** 3 good
**Presentation:** 4 excellent
**Contribution:** 4 excellent
**Rating:** 8
**Confidence:** 5

**Summary:**

This paper introduces Pick-a-pic, a large and open dataset of human preference over model-generated images. Using this dataset, a scoring function (PickScore) based on CLIP is trained. The paper finds that PickScore has better correlation with real user preferences compare to existing scoring methods. Furthermore, PickScore can be used as an output selection method to improve the generated image's quality.

**Strengths:**

- The proposed dataset fills an important gap for image generation research. Human preference is an important metric and should be considered when evaluating image generation models.

- The dataset is collected from real user, contains a large number examples, and most importantly, is open-sourced.

- The learned PickScore shows high correlation with real user preference. Although expected from training on Pick-a-pic, it still provides a valuable alternative over existing evaluation metrics.

- It is nice to see the potential of PickScore being used to develop better image generation models.

**Weaknesses:**

- It would be good if the paper could provide a more comprehensive analysis of the prompts used by the users in Pick-a-pic. How diverse are the prompt? What is the word distribution difference between Pick-a-pic and existing datasets such as COCO or DreamBench?

- It is possible that the preference of general public users could be biased towards certain aspects of the generated image such as its interestingness, while neglecting other aspects such as faithfulness to the prompt. It would be good if the authors could provide some analysis on the specific types of biases observed in Pick-a-pic.



**Questions:**

- Is it possible to ensemble PickScore with other scoring functions as a more comprehensive evaluation / model selection method?

**Limitations:**

It would be good to see a more detailed analysis on the limitations of the dataset and PickScore.

---

> ### Author Rebuttal · Authors · 2023-08-05
>
> Weaknesses:
> 1. It would be good if the paper could provide a more comprehensive analysis of the prompts used by the users in Pick-a-pic. How diverse are the prompt? What is the word distribution difference between Pick-a-pic and existing datasets such as COCO or DreamBench?
>
> Answer: We agree that helping readers and potential users of the dataset to better understand the distribution of prompts of Pick a Pic when compared to existing benchmarks can be insightful. We will gladly add to the paper examples of randomly sampled examples from Pick a Pic, alongside prompts from existing benchmarks. We will also consider visualizing word distributions. However, we usually find them to be less intuitive and interpretable for readers. Here are randomly sampled examples from Pick a Pic, and MS-COCO (which involved day-to-day scenes, that are probably less relevant for text-to-image users):
>
> Pick a Pic -  "forest with ruins, photo"	, “A panda bear as a mad scientist”, "product photo of a sneakers", “photo of a bicycle, detailed, 8k uhd, dslr, high quality, film grain, Fujifilm XT3”, “female portrait photo”, “Alexander the great, cover art, colorful”, "A galactic eldritch squid towering over the planet Earth, stars, galaxies and nebulas in the background…”, "Portrait of a giant, fluffy, ninja teddy bear”, "insanely detailed portrait, dartH vader, shiny, extremely intricate, high res, 8k, award winning", "Giant ice cream cone melting and creating a river through a city"
>
> MS-COCO - "A man with a red helmet on a small moped on a dirt road. ", "A woman wearing a net on her head cutting a cake. ", "there is a woman that is cutting a white cake", "a little boy wearing headphones and looking at a computer monitor", "A young girl is preparing to blow out her candle.”, "A commercial stainless kitchen with a pot of food cooking. ", "Two men that are standing in a kitchen.", "A man riding a bike past a train traveling along tracks.", "The pantry door of the small kitchen is closed.", "A man is doing a trick on a skateboard"
>
> 2. It is possible that the preference of general public users could be biased towards certain aspects of the generated image such as its interestingness, while neglecting other aspects such as faithfulness to the prompt. It would be good if the authors could provide some analysis on the specific types of biases observed in Pick-a-pic.
>
> Answer: We *strongly* agree that interpreting which attributes (like interestingness, artistic flavor, etc.) affect human preferences is an interesting question. However, we are afraid that conducting a comprehensive and informed user study on this topic may require expertise in human–computer interaction. Therefore, we will clarify in the limitations section that while we did not explore it in the context of our work, we hope that the available dataset and other artifacts of this work will enable the community to study these phenomena rigorously.
>
> Questions:
> 1. Is it possible to ensemble PickScore with other scoring functions as a more comprehensive evaluation / model selection method?
>
> Answer: This is a very interesting question and we will try to answer this question from different perspectives. We believe that the best practice will be to report results of various scoring functions (for example PickScore, CLIPScore, and aesthetic predictor). However, we think that human preference is the most important metric to optimise, and hence, ensembling it with proxies for human preference (e.g. text alignment or aesthetics), may stir the ensembled metric from human preferences. Finally, we hope that the community will follow by collecting and open-sourcing human preference data which will allow researchers to train more robust and updated scoring functions.

---

> > ### Comment · Reviewer_hbT1 · 2023-08-11
> >
> > I appreciate the authors' response. I confirm my rating of strong accept.

---

### Official Review · Reviewer_UW8i · 2023-07-02

**Soundness:** 4 excellent
**Presentation:** 4 excellent
**Contribution:** 3 good
**Rating:** 7
**Confidence:** 4

**Summary:**

This paper introduces a dataset of text-to-image prompts, along with user preferences over the generated images. This is done by building a web application where multiple images are generated for a given prompt, and the user is asked to select their preferred image. As a result of collecting this Pick-a-Pic dataset, a CLIP model (with a ViT-H backbone) is finetuned on these user preferences to obtain a scoring function PickScore that demonstrates strong results on evaluating text-to-image alignment.  The results indicate that the proposed model PickScore is not only able to significantly outperform standard text-image alignment metrics (i.e CLIPScore), but also performs better than some contemporary work for the same. Additionally, the paper also demonstrates that selecting the best image from a large set of images generated by varying the prompts and seed using PickScore results in images that are preferred over other selection strategies.

**Strengths:**

Overall, I think the paper makes a very solid and timely contribution.
1) Existing methods for evaluating text-to-image alignment have been quite limited, and therefore a large-scale dataset of user preferences that is crowdsourced would be very valuable.
2) Furthermore, the PickScore model can be useful for a variety of downstream applications since it seems to be atleast on par with humans in evaluating text-image alignment.
3) The experiments in the paper are also quite thorough and have comparisons to a lot of different models for evaluating text-to-image, including contemporary work.

**Weaknesses:**

I believe that the paper tackles a problem that is of interest to sufficient audience, and achieves all/most of the claims made in the paper, therefore I do not have major weaknesses to discuss. Naturally, there could be alternative methods/objectives to train better models for improved performance, different vision-language models (i.e BLIP/BLIP-2) could have been considered, or using PickScore to improve text-to-image generation models could all be done in future work.

**Questions:**

My only major question is regarding the "superhuman" performance achieved by the PickScore model. Given that the ground truth is captured from human preferences, I do not fully understand what achieving superhuman performance on this benchmark means (and the description in the paper is quite short lines 176-178). A slightly more detailed explanation of this would be very helpful.

**Limitations:**

Yes, I think the limitations section in the paper covers the major issues that might arise from this work (i.e dataset having NSFW images, some noise in cases where the users of the applications make mistakes). Apart from this, the paper also mentions that they have not run a hyperparameter sweep, so it is possible for some future work to train an improved model using the Pick-a-Pic dataset.

---

> ### Author Rebuttal · Authors · 2023-08-05
>
> Questions:
>
> 1. My only major question is regarding the "superhuman" performance achieved by the PickScore model. Given that the ground truth is captured from human preferences, I do not fully understand what achieving superhuman performance on this benchmark means (and the description in the paper is quite short lines 176-178). A slightly more detailed explanation of this would be very helpful.
>
> Answer: We will clarify and elaborate on this question in the paper. There is a core and very important difference between *users* that produce the ground truth and *annotators* used to evaluate human performance. The *users* that produce the ground truth are actual text-to-image users, which have a context, and an idea (which may be incomplete) for an image, and invent a prompt (which may lack details) with hope that the resulting image will match their preferences. In contrast, *annotators* that are used to measure human performance are oblivious to the user’s context, idea, and motivation. Therefore, superhuman performance on this task means that the model is able to outperform a human *annotator* that is oblivious to the original *user’s* context, idea, and motivation.

---

> > ### Comment · Reviewer_UW8i · 2023-08-18
> >
> > Thanks for the clarification. Having gone through all the reviews, and the response provided by the authors, I am happy to stick with my rating (Accept).

---

### Official Review · Reviewer_UkC5 · 2023-07-04

**Soundness:** 3 good
**Presentation:** 3 good
**Contribution:** 3 good
**Rating:** 6
**Confidence:** 5

**Summary:**

The study gathers a dataset of user preferences by soliciting human choices regarding more preferred images in response to text prompts given a text prompt. Besides, the authors propose a metric called PickScore by fine-tuning CLIP-H on this dataset. The experiments show that PickScore demonstrates better alignment with human judgment compared to existing evaluation metrics such as FID or HPS.

**Strengths:**

In this paper, a novel dataset named Pick-a-Pic is constructed, considering human preferences. This dataset proves to be valuable as it can be used to train an automatic evaluation metric, such as the proposed PickScore, to effectively assess the quality of text-to-image models.

**Weaknesses:**

1. It is important to acknowledge that the proposed Pick-a-Pic dataset may exhibit certain biases due to the usage of images generated by existing generative models. As a result, the PickScore metric, trained on this dataset, could potentially inherit the same biases and limitations.
2. Some experimental results are confused, e.g., the correlation value between FID and human judgment is -0.9, which approaches the lower bound of correlation. Does it mean there is a negative correlation between FID and human judgment?

Please refer to more details in “Question” below.

**Questions:**

*Dataset and Method

The concerns regarding dataset biases and the potential biases associated with PickScore.

(1) Considering that PickScore is trained on high-quality images, there is a concern about its ability to accurately evaluate generative models that produce comparatively lower-quality images.

(2) Since the images in the Pick-a-Pic dataset are generated by models such as Stable Diffusion 2.1, there is a possibility that PickScore could exhibit bias towards these models, potentially resulting in higher score values.

*Experiment

(1) The PickScore metric is trained on the proposed Pick-a-Pic dataset, which is annotated by humans. However, it is intriguing to observe that it surpasses human performance in Table 1(b).

(2) Confusion about the results of FID.

1. The reason behind the correlation value of -0.9 between FID and human judgment, which nearly reaches the lower bound of correlation, remains unclear. Although the paper claims the reason may lie in “larger scales tend to produce more vivid images (which humans typically prefer), but differ from the distribution of ground truth images in MS-COCO, yielding worse (higher) FID scores”, why the more vivid image would be totally different from the distribution of GT images on MS-COCO? They are even negatively correlated.

2. Which feature extractor is employed in this FID calculation? Is it an ImageNet pre-trained model? If so, is it suitable for capturing the features of the aesthetic or surreal images on the proposed dataset?
(3) How about the performance of PickScore on realistic images, it seems most of the images on the proposed dataset belong to aesthetic or surreal ones.

*Minor Issues

Line 173: Figure 1a -> Table 1a

**Limitations:**

This paper mentions the limitations and broader impact in Section 8: “some images and prompts may contain NSFW content that could potentially bias the data and some users may have made judgments without due care”. It would be better to double- or triple-check each sample on the dataset by humans, as the quality of data would greatly affect its usability and reliability.

---

> ### Author Rebuttal · Authors · 2023-08-05
>
> Weaknesses:
>
> 1. Some experimental results are confused, e.g., the correlation value between FID and human judgment is -0.9, which approaches the lower bound of correlation. Does it mean there is a negative correlation between FID and human judgment?
>
> Answer: Regarding the specific example and question if our results indicate that there is a negative correlation between FID and human judgment - the answer is YES, and we *explicitly* discuss this in the paper (e.g. paragraph in lines 216-224). Moreover, this finding aligns with the fact that the vast majority of the community generates images with high classifier free guidance scale values (e.g. 7) when generating images, which tend to produce much worse FID scores, while abstain from using low classifier free guidance scale values (e.g. 3), that produce better FID scores. It is unclear to us which other experimental results you find to be confusing.
>
> Questions:
>
> 1. Considering that PickScore is trained on high-quality images, there is a concern about its ability to accurately evaluate generative models that produce comparatively lower-quality images.
>
> Answer: While we agree that PickScore’s ability to evaluate models that produce comparatively lower-quality images has not been thoroughly tested, it is unclear whether it should be a concern. Currently, the standard practice for evaluating new models is via FID against MS-COCO. We show that this protocol is lacking, suggest a new protocol, and back its utility with strong empirical evidence. We hope that our paper will lead the community to suggest improved protocols, use our data collection methodology, dataset, model, evaluation procedure (e.g. use of Elo ratings), code for training the model, etc. as basis for such future work.
>
> 2. Since the images in the Pick-a-Pic dataset are generated by models such as Stable Diffusion 2.1, there is a possibility that PickScore could exhibit bias towards these models, potentially resulting in higher score values.
>
> Answer: We have included different models in the Pick a Pic dataset (e.g. SD2.1, DP2.0, and many SDXL early variants), and randomized classifier free guidance scales - which overall include *dozens* of data points (please see Fig 6, and lines 229-230). We agree that as new models are created it is important to assert that PickScore’s performance remains high. Moreover, we believe that if empirical evidence will show that PickScore requires an “update”, the foundations that we have laid in this paper, will prove to be very valuable.
>
> 3. The PickScore metric is trained on the proposed Pick-a-Pic dataset, which is annotated by humans. However, it is intriguing to observe that it surpasses human performance in Table 1(b).
>
> Answer: We will clarify and elaborate on this question in the paper. There is a core and very important difference between *users* that produce the ground truth and *annotators* used to evaluate human performance. The *users* that produce the ground truth are actual text-to-image users, which have a context, and an idea (which may be incomplete) for an image, and invent a prompt (which may lack details) with hope that the resulting image will match their preferences. In contrast, *annotators* that are used to measure human performance are oblivious to the user’s context, idea, and motivation. Therefore, superhuman performance on this task means that the model is able to outperform a human *annotator* that is oblivious to the original *user’s* context, idea, and motivation.
>
> 4. The reason behind the correlation value of -0.9 between FID and human judgment, which nearly reaches the lower bound of correlation, remains unclear. Although the paper claims the reason may lie in “larger scales tend to produce more vivid images (which humans typically prefer), but differ from the distribution of ground truth images in MS-COCO, yielding worse (higher) FID scores”, why the more vivid image would be totally different from the distribution of GT images on MS-COCO? They are even negatively correlated.
>
> Answer: MS-COCO images were taken by amateur photographers in day-to-day settings. Such images tend to be more ordinary and less vivid than images that humans wish to see when generating images. Our hypothesis is that humans wish to see images taken by professional photographers or artists for example.
>
> 5. Which feature extractor is employed in this FID calculation? Is it an ImageNet pre-trained model? If so, is it suitable for capturing the features of the aesthetic or surreal images on the proposed dataset?
>
> Answer: As we have mentioned in the paper, we have used the *standard practice* for FID calculation (similarly to that used in Make-a-Scene, DALLE2, Imagen, etc.) - specifically and similarly to many other papers, we have used the script from https://github.com/MinfengZhu/DM-GAN/blob/master/eval/FID/fid_score.py . One of the contributions of our paper is advocating that the *standard practice* of conducting FID evaluation against MS-COCO does not align well with text-to-image user preferences, and therefore suggest a new protocol for model evaluation.
>
> 6. How about the performance of PickScore on realistic images, it seems most of the images on the proposed dataset belong to aesthetic or surreal ones.
>
> Answer: One of the benefits of allowing users to specify their prompts, is that it enables us to acquire *relevant* and *diverse* prompts, *many* of which are realistic. Additionally, we also conducted experiments that exclusively use realistic images from MS-COCO and witnessed much better correlation than the standard practice of using FID.

---

> > ### Comment · Area_Chair_N6Le · 2023-08-19
> >
> > Dear authors,
> >
> > Thank you for taking the time to address the reviewer's comments.
> >
> > Dear reviewer UkC5,
> >
> > Do you have any additional thoughts or concerns after reading the response?
> >
> > Best,
> >
> > AC

---

> > > ### Comment · Reviewer_UkC5 · 2023-08-19
> > >
> > > Thanks for the detailed response, which addressed all of my concerns. I will raise the score.

---

### Official Review · Reviewer_c5Um · 2023-07-06

**Soundness:** 3 good
**Presentation:** 3 good
**Contribution:** 3 good
**Rating:** 6
**Confidence:** 4

**Summary:**

This paper contributes a new dataset of human preferences for text-to-image (T2I) generations. It then trains a CLIP-based score function to that can predict human preferences between two generated images. The paper then shows how this can be used to better align a T2I model to human preferences.

**Strengths:**

-- Research angle is very timely on better aligning a T2I model to human preferences.

-- The proposed dataset and the learned CLIP-based human image preference function can benefit a diverse cohort of vision researchers.

-- The paper is overall easy to read.

**Weaknesses:**

While the proposed dataset itself is a significant enough contribution towards acceptance, the paper's technical contribution is relatively trivial and calls for further discussion:

-- the way how the paper leverage a model to learn the human preferences; the paper fine-tunes the entire weights of a pre-trained CLIP and leverages KL loss function to align it to a binary human decision. Given parameter-efficient fine-tuning (PEFT) approach such like LoRA or prompt/adaptor learning has shown exception promise and the fact that the paper only collects a moderate scale of human preferences (less then a million), full fine-tuning might be sub-optimal. The KL loss function choice also needs further ablations, does regression to human preference work?

-- On the held-out validation set, the paper's final results are merely around 70% of the time that align with human preference. Given the random chance is $\sim$ 55\%, this is not a particularly good number. The paper needs more extensive discussion to elaborate on this case; is it due to the over noisy human preference ratings that model can not find a globally cohesive way to distil an understanding or the the proposed model is not sophisticated enough to carve out the necessary information for learning?

-- The collected human preference ratings are based on synthetic images. What is their impact towards real images? Can the developed human preference scoring function be used as a plug-n-play model that benefits computer vision tasks with need for better human alignment?

**Questions:**

Please see weaknesses.

---

> ### Author Rebuttal · Authors · 2023-08-05
>
> Weaknesses:
>
> 1. While the proposed dataset itself is a significant enough contribution towards acceptance, the paper's technical contribution is relatively trivial and calls for further discussion.
>
> Answer: While we appreciate the reviewer's note that the dataset itself is a significant enough contribution towards acceptance, we do not agree that the paper's technical contribution is limited. We have acquired state-of-the-art scoring function (PickScore) by applying and adapting best practices from preference modeling in NLP into that of T2I and showed its effectiveness in different scenarios, such as preference prediction, model evaluation, and improving text-to-image models via ranking. We believe that each scenario is important as it can pave the way for future publications, and prove to be valuable to the research community.
>
> Questions:
>
> 1. the paper fine-tunes the entire weights of a pre-trained CLIP and leverages KL loss function to align it to a binary human decision. Given parameter-efficient fine-tuning (PEFT) approach such like LoRA or prompt/adaptor learning has shown exception promise and the fact that the paper only collects a moderate scale of human preferences (less then a million), full fine-tuning might be sub-optimal.
>
> Answer: LoRA is a great way to reduce memory requirements while training, but we are not familiar with evidence that it should result in better downstream performance - especially given a substantial dataset like Pick-a-Pic that has more than half-a-million examples. Hence, we are not sure how the use of LoRA may contribute to the research questions that we have investigated, and therefore, we did not experiment with it.
>
> 2. The KL loss function choice also needs further ablations, does regression to human preference work?
>
> Answer: Our data consists of relative rankings, and it is therefore very natural to solve this problem with KL loss. Please refer to extensive work of human-preference modelling in NLP e.g. InstructGPT, that we have considered when we made this choice. Specifically, regression is an inadequate function for this learning problem (and therefore, is also not being used in other papers that investigate human-preference modeling over relative preferences). For example, if we have several data points that correspond to the same prompt with images A,B,C,D and human preference that label, A < B = C < D - it is unclear which global value each image should get in the regression problem.
>
> 3. On the held-out validation set, the paper's final results are merely around 70% of the time that align with human preference. Given the random chance is 55%, this is not a particularly good number. The paper needs more extensive discussion to elaborate on this case; is it due to the over noisy human preference ratings that model can not find a globally cohesive way to distil an understanding or the the proposed model is not sophisticated enough to carve out the necessary information for learning?
>
> Answer: We will gladly elaborate more in the paper about this question. Concretely, we show that humans' ability to predict user preferences is *even more limited* than PickScore. Hence, we can conclude (1) PickScore performance can be considered to be quite good, and (2) the difficulty resides (entirely or partially) with the data itself, rather than the model’s learning capacity.
>
> 4. The collected human preference ratings are based on synthetic images. What is their impact towards real images?
>
> Answer: Can you please clarify what do you mean "What is their impact towards real images?"?
>
> 5. Can the developed human preference scoring function be used as a plug-n-play model that benefits computer vision tasks with need for better human alignment?
>
> Answer: Do you refer to the ranking experiment in which we have showed the benefits of using PickScore to improve vanilla text-to-image generation? We would appreciate a clarification to correctly answer this question.

---

> > ### Comment · Area_Chair_N6Le · 2023-08-19
> >
> > Dear authors,
> >
> > Thank you for taking the time to address the reviewer's comments.
> >
> > Dear reviewer c5Um,
> >
> > Do you have any additional thoughts or concerns after reading the response?
> >
> > Best,
> >
> > AC

---

### Author Rebuttal · Authors · 2023-08-05

We thank the reviewers for their thoughtful feedback. We are encouraged that they found our paper to be a very solid (UW8i), timely (UW8i, c5Um), and significant (c5Um) contribution, and our experiments to be thorough (UW8i). We are glad that they consider our open-sourced and large dataset of human (specifically - *real users*)   preference in text-to-image, and learned human-preference scoring function (PickScore) to be very valuable (UW8i, UkC5), benefit a diverse cohort of researchers (c5Um), and prove to be useful for a variety of downstream applications (UW8i, hbT1), as it fills an important gap in image generation research (hbT1). We address each of the reviewers' feedback on weaknesses and questions in the corresponding dedicated rebuttal section.

---

### Decision · Program_Chairs · 2023-09-21

**Decision:**

Accept (poster)

**Comment:**

This paper proposes a human preference dataset for text-to-image generation and provides a reward function called PickScore.

**strengths**

* Large-human preference dataset for text-to-image generation
* Reward model which can be useful for various downstream tasks
* Solid experimental results

**weaknesses/suggestions**

* Lack of discussions on potential bias in human dataset
* Evaluating the quality of reward model on real images.

Overall, the reviewers and AC agree that this paper makes a worthy contribution to NeurIPS.
However, I encourage the authors to think carefully about how to reflect the comments or resolve the questions from reviewers in the camera ready version.